# One-sample Guided Object Representation Disassembling

**Zunlei Feng**
Zhejiang University
zunleifeng@zju.edu.cn

**Yongming He**
Zhejiang University
yongminghe@zju.edu.cn

**Xinchao Wang**
Stevens Institute of Technology
xinchao.wang@stevens.edu

**Xin Gao**
Alibaba Group
zimu.gx@alibaba-inc.com

**Jie Lei**
Zhejiang University of Technology
jasonlei@zjut.edu.cn

**Cheng Jin**
Fudan University
jc@fudan.edu.cn

**Mingli Song**$^*$
Zhejiang University
brooksong@zju.edu.cn

## Abstract

The ability to disassemble the features of objects and background is crucial for many machine learning tasks, including image classification, image editing, visual concepts learning, and so on. However, existing (semi-)supervised methods all need a large amount of annotated samples, while unsupervised methods can't handle real-world images with complicated backgrounds. In this paper, we introduce the One-sample Guided Object Representation Disassembling (One-GORD) method, which only requires one annotated sample for each object category to learn disassembled object representation from unannotated images. For the annotated one-sample, we first adopt some data augmentation strategies to generate some synthetic samples, which can guide the disassembling of the object features and background features. For the unannotated images, two self-supervised mechanisms: dual-swapping and fuzzy classification are introduced to disassemble object features from the background with the guidance of annotated one-sample. What's more, we devise two metrics to evaluate the disassembling performance from the perspective of representation and image, respectively. Experiments demonstrate that the One-GORD achieves competitive dissembling performance and can handle natural scenes with complicated backgrounds.

## 1 Introduction

Learning disassembled object representation is a vital step in many machine learning tasks, including image editing, image classification, few/zero-shot learning, and visual concepts learning. For example, many image editing [4, 21, 26] for objects typically rely on image segmentation techniques and human labor, which only handles object in image level. The existing classification works [15, 18] usually train classifiers with large amounts of annotated samples to extract specific object features and identify them, which also has a serious cost of labor, time, and memory. For the few/zero-shot learning problem, most of works [2, 22, 27, 30] adopt representations extracted by pre-trained deep models as the features of specific objects. However, the representations extracted by pre-trained models usually contain many irrelevant features, which will disturb the performance of models. So, an object representation learning method that can learn the pure and entire features of the specific object with a few annotated data is desperately needed.

---

$^*$Corresponding author.

Until now, most object representation learning methods [6, 8, 12, 13, 23] are proposed to handle simple scenes with multiple objects in an unsupervised manner. However, those methods can't handle real-world images with complicated backgrounds, which limits their application in many machine learning tasks. On the other hand, the existing supervised object representation learning methods are rarer. Some (semi-)supervised disentangling methods [11, 25] can be transferred to learn disassembled object representation through annotating the object information as labels. However, it still requires many annotated samples. Another line of works [14, 28] is concerned with obtaining the segmentation of objects and does not learn structured object representations.

In this paper, we propose the One-sample Guided Object Representation Disassembling (One-GORD) method, which only requires one annotated sample for each object category to learn disassembled object representation from a large number of unannotated images. The proposed One-GORD is composed of two modules: the augmented one-sample supervision module and the guided self-supervision module. In the one-sample guided module, we first generate some synthetic sample pairs with data augmentation strategies. Then, following the "encoding-swapping-decoding" architecture, we swap the parts of their representations to reconstruct the synthetic ground-truth pairs, which guides the features of objects and backgrounds to be decoded into different parts of the representations.

In the guided self-supervision module, we introduce two self-supervised mechanisms: fuzzy classification and dual swapping. For the dual swapping, given a pair of samples that are composed of the annotated one-sample and an unannotated image, we swap the first halves of their representations to reconstruct the hybrid images. Then, the first halves of the hybrids' representations are swap back to reconstruct the original pair samples, which formats the self-supervision loss for the disassembling of unannotated images with the guidance of annotated one-sample. Meanwhile, the fuzzy classification supervises the first and latter halves of representation to extract features of any object category and background, respectively.

Furthermore, to verify the effectiveness of the proposed method, we devise two metrics to evaluate the modularity of representations and the integrity of images. The former measures the modularity and portability of the latent representations, while the latter evaluates the visual completeness of the reconstructed images. As will be demonstrated in our experiments, the proposed One-GORD achieves truly promising performance.

Our contribution is the proposed One-GORD, which only requires one annotated sample for learning disassembled object representation. Two self-supervised mechanisms format self-supervised losses for the disassembling of unannotated image representations with the guidance of annotated one-sample. Meanwhile, We also introduce two disassembling metrics, upon which the proposed One-GORD achieves truly encouraging results.

## 2    Related Work

Representation learning [5] has achieved several breakthroughs. This includes representation disentangling [7, 16, 17], which disentangle attribute features into different parts of representation. Part of (semi-)supervised disentangling methods [25, 11] can be transferred to learn disassembled object representation through annotating the object information as labels. However, it still requires a lot of annotated samples. Another line of works is concerned with obtaining the segmentation of objects without considering representation learning. Most current approaches [28, 14] require explicitly annotated segmentations in the dataset, which limits the generalization of these models. Furthermore, these methods typically only segment images and don't learn structured object representations.

Works for learning disassembled object representation are relatively rarer. Burgess *et al.* [6] proposed the MONet, where a VAE is trained end-to-end together with a recurrent attention network to provide attention masks around regions of images. The MONet can decompose 3D scenes into objects and background elements. Greff *et al.* [12] developed an amortized iterative refinement based method, which can segment scenes and learn disentangled object features. Van Steenkiste *et al.* [23] proposed the R-NEM that learns to discover objects and model their physical interactions from raw visual images, which is the extension of N-EM [13]. Dittadi and Winther [8] proposed the probabilistic generative model for learning of structured, compositional, object-based representations of visual scenes. Engelcke *et al.* [10] proposed the GENESIS for rendered 3D scenes, which decomposes and generates scenes by capturing relationships between scene components. Lin *et al.* [20] proposed the SPACE, which factorizes object representations of foreground objects while also decomposing

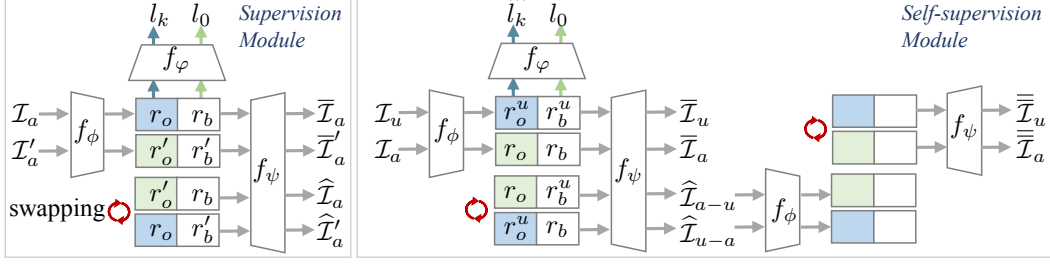

Figure 1: The architecture of the proposed One-GORD. It comprises two modules: the augmented one-sample supervision module and the guided self-supervision module. The former one is employed for the augmented annotated one-samples $(\mathcal{I}_a, \mathcal{I}'_a)$ while the latter is for unannotated image $\mathcal{I}_u$ and the annotated image $\mathcal{I}_a$. $f_\phi$, $f_\psi$ and $f_\varphi$ denote the encoder, decoder, and classifier, respectively. $l_0$, $l_k$ and $\overset{...}{l}_k$ denote the background label, the $k$-th object category label and the $k$-th unknown object category label. $\overline{\mathcal{I}}_a$, $\overline{\mathcal{I}}'_a$ and $\overline{\mathcal{I}}_u$ denotes the reconstructed images. $\widehat{\mathcal{I}}_a$, $\widehat{\mathcal{I}}'_a$, $\widehat{\mathcal{I}}_{a-u}$ and $\widehat{\mathcal{I}}_{u-a}$ denote the hybrid images. $\overline{\overline{\mathcal{I}}}_u$ and $\overline{\overline{\mathcal{I}}}_a$ denote the dual reconstructed images.

background segments of complex morphology. All the above methods learn disassembled object representation in an unsupervised manner. However, those methods work on synthetic images only but not real-world ones.

## 3 The Proposed Method

In this section, we give more details of our proposed One-GORD model (Fig. 1). We start by introducing the augmented one-sample supervision module, which disassembles the object features from the background with the supervision information of augmented annotated one-samples. Then, we describe the self-supervision module, which disassembles the object representation with two self-supervised mechanisms under the guidance of the annotated one-sample. Finally, we summarize the complete algorithm.

### 3.1 Augmented One-sample Supervision Module

**Annotated One-sample** The natural scene images are usually composed of complicated things, which results in that unannotated object representation leaning methods can't work well. To distinguish the foreground objects from the background, there must be one annotated sample for each object category that is intended to be disassembled. In this paper, we choose one image for each object category and annotate the category label $l_k, k \in \{1, 2, 3, ..., K\}$ and mask of the object, where $K$ is the number of object categories. To enhance the influence of one-samples' supervision, some data augmentation strategies are adopted to generated augmented images. The augmentation strategies include mirroring, adding noise, and changing background, which are usually optional for different datasets. For the augmented images, we randomly choose two samples $\mathcal{I}_a$ and $\mathcal{I}'_a$, then get the ground-truth images $\widetilde{\mathcal{I}}_a$ and $\widetilde{\mathcal{I}}'_a$ by swapping their objects with the masks.

**Supervision Disassembling** The supervision information contains two parts: *classification supervision* and *reconstruction supervision*. With encoder $f_\phi$ and decoder $f_\psi$, the input images $\mathcal{I}_a$ and $\mathcal{I}'_a$ are encoded into the representations $R_a$ and $R'_a$, which are decoded into the images $\overline{\mathcal{I}}_a$ and $\overline{\mathcal{I}}'_a$. The latent representations are constrained to contain all the features of input images with the following basic reconstruction loss $\mathcal{L}_{\mathbf{rec}}$:

$$\mathcal{L}_{\mathbf{rec}} = ||\mathcal{I}_a - \overline{\mathcal{I}}_a||_2^2 + ||\mathcal{I}'_a - \overline{\mathcal{I}}'_a||_2^2.$$

Then, the representations $R_a$ and $R'_a$ are divided into two parts: $[r_o, r_b]$ and $[r'_o, r'_b]$, respectively.

For the *classification supervision*, the object label $l_k, k \in \{1, 2, 3, ..., K\}$ supervises $r_o$ and $r'_o$ to extract object features while the background label $l_0$ supervises $r_b$ and $r'_b$ extracting background

features with the classification loss $\mathcal{L}_{\mathbf{cla}}$:

$$\mathcal{L}_{\mathbf{cla}} = -\sum_{k=0}^{K}\sum^{vec} l_k \times \log(p),$$

where $l_k$ is a one-hot label vector, $p$ is the predicted probability of classifier $f_\varphi$ with one part of the representations $\{r_o, r_b, r'_o, r'_b\}$ as the input, $\sum^{vec}$ denotes the summation of $n$-dimension vector.

For the *reconstruction supervision*, through swapping the object parts $r_o$ and $r'_o$, the hybrid representations $[r'_o, r_b]$ and $[r_o, r'_b]$ are decoded into the hybrid images $\widehat{\mathcal{I}}_a$ and $\widehat{\mathcal{I}}'_a$, respectively. The ground-truth images $\widetilde{\mathcal{I}}_a$ and $\widetilde{\mathcal{I}}'_a$ supervise the first half and latter half to extract features of object and background with the following reconstruction supervision loss $\mathcal{L}'_{\mathbf{rec}}$:

$$\mathcal{L}'_{\mathbf{rec}} = ||\widetilde{\mathcal{I}}_a - \widehat{\mathcal{I}}_a||_2^2 + ||\widetilde{\mathcal{I}}'_a - \widehat{\mathcal{I}}'_a||_2^2.$$

## 3.2   Guided Self-supervision Module

For large amounts of unannotated images, we introduce two self-supervised mechanisms: dual swapping and fuzzy classification. Dual swapping swaps parts of the unannotated image representation back and forth to reconstruct the original image, which generates the self-supervised information. The fuzzy classification supervises the features of unknown objects and background to be encoded into different parts of the representations with the fuzzy classification loss.

**Dual Swapping**   For the unannotated image $\mathcal{I}_u$, the same autoencoder reconstructs it as image $\overline{\mathcal{I}}_u$ with the following unsupervised reconstruction loss $\mathcal{L}^{\mathbf{u}}_{\mathbf{rec}}$:

$$\mathcal{L}^{\mathbf{u}}_{\mathbf{rec}} = ||\overline{\mathcal{I}}_u - \mathcal{I}_u||_2^2.$$

Similarly, the encoded representation $R_u = f_\phi(\mathcal{I}_u)$ is divided into two parts $[r_o^u, r_b^u]$. To bring the guidance of annotated one-samples, we swap the representations' first parts of annotated one-samples and unannotated images. Then, the hybrid representations $[r_o^u, r_b]$ and $[r_o, r_b^u]$ are decoded into the hybrid images $\widehat{\mathcal{I}}_{a-u}$ and $\widehat{\mathcal{I}}_{u-a}$. Following the "encoding-swapping-decoding" process again, the hybrid images $\widehat{\mathcal{I}}_{a-u}$ and $\widehat{\mathcal{I}}_{u-a}$ are reconstructed into $\overline{\overline{\mathcal{I}}}_u$ and $\overline{\overline{\mathcal{I}}}_a$ by swapping the representations' first parts back. If the representation is well disassembled, the dual reconstructed image $\overline{\overline{\mathcal{I}}}_u$ should reconstruct the original image $\mathcal{I}_u$. So, the dual swapping loss $\mathcal{L}^{\mathbf{d}}_{\mathbf{rec}}$ is defined as follows:

$$\mathcal{L}^{\mathbf{d}}_{\mathbf{rec}} = ||\overline{\overline{\mathcal{I}}}_u - \mathcal{I}_u||_2^2.$$

Meanwhile, to ensure that the object features are encoded into the first part of one-sample, the hybrid image $\widehat{\mathcal{I}}_{a-u}$ should have the same object with the annotated one-sample $\mathcal{I}_a$. So, the object reconstruction loss $\mathcal{L}^{\mathbf{o}}_{\mathbf{rec}}$ is defined as follows:

$$\mathcal{L}^{\mathbf{o}}_{\mathbf{rec}} = \mathcal{M}_a \times ||\widehat{\mathcal{I}}_{a-u} - \mathcal{I}_a||_2^2,$$

where $\mathcal{M}_a$ is the object mask of the one-sample $\mathcal{I}_a$. The interaction between the unannotated images and annotated images will enhance the guidance of the annotated one-sample.

**Fuzzy Classification**   For the unannotated images, the object labels are unknown. It's hard to supervise the fixed part to extract specific object features. Nevertheless, the object features of the unannotated images are still discriminative and the features of the background should be different from them. If the first half of the representation containing pure object features, it will be classified into the particular object category easily. So we devise the fuzzy classification loss, which can constrain that the features of unknown objects be classified into their original categories. What's more, the fixed label $l_0$ is also adopted to differentiate the background features from object features. So, fuzzy classification loss $\mathcal{L}^{\mathbf{z}}_{\mathbf{rec}}$ is defined as follows:

$$\mathcal{L}^{\mathbf{z}}_{\mathbf{cla}} = -\log\{[1 - \sum^{vec}\prod_{k=1}^{K}(l_k - l_k \times q)]\} - \tau\sum^{vec} l_0 \times \log(q_0),$$

where $l_k$ is one-hot object label, $q = f_\varphi(r_o^u)$ is the predicted probability of classifier $f_\varphi$ with the first half of the representation $r_o^u$ as input, $q_0 = f_\varphi(r_b^u)$ is the predicted probability of classifier $f_\varphi$ with the latter half of the representation $r_b^u$ as input, $\sum^{vec}$ denotes summation of multi-dimension vector, and $\tau$ is the balance parameter.

### 3.3 Complete Algorithm

In summary, the total loss $\mathcal{L}$ contains all the loss terms in the above two modules. In the supervision module, the loss terms disassemble the object representation with the classification supervision and reconstruction supervision; In the self-supervision module, the loss terms disassemble the object representation with two self-supervision mechanisms under the guidance of the annotated one-samples. The total loss $\mathcal{L}$ is given as follows:

$$\mathcal{L} = \alpha\mathcal{L}_{\mathbf{rec}} + \beta\mathcal{L}_{\mathbf{cla}} + \gamma\mathcal{L}'_{\mathbf{rec}} + \eta\mathcal{L}^{\mathbf{u}}_{\mathbf{rec}} + \lambda\mathcal{L}^{\mathbf{d}}_{\mathbf{rec}} + \rho\mathcal{L}^{\mathbf{o}}_{\mathbf{rec}} + \delta\mathcal{L}^{\mathbf{z}}_{\mathbf{cla}},$$

where $\alpha$, $\beta$, $\gamma$, $\eta$, $\lambda$, $\rho$, and $\delta$ are the balance parameters. It is noticeable that all the encoders, decoders, and classifiers share the same parameters, respectively.

## 4 Disassembling Metric

It's essential to measure the disassembling performance of different methods. To the very best of our knowledge, there is no quantitative metric for evaluating the disassembling performance directly. To measure the disassembling performance fairly, we begin by defining the properties that we expect the disassembled object representation to have. It should consider the latent representation and the reconstructed image. If the object representation is disassembled perfectly, the extracted object representation should be equally for different images with the same object. On the other hand, for the images reconstructed with the same object representation, objects should keep the same.

Therefore, we devise two disassembling metrics to measure the *modularity* on the latent representation and the *integrity* of the reconstructed image, respectively. For *modularity*, we run inference on images that contain the fixing object and different backgrounds. If the modularity property holds for the inferred representations, there will be less variance in the inferred latent representations that correspond to the fixed object. For the $T * D$ test images that are composed of $T$ kinds of object and $D$ image for each object category, the *Modularity Score* $M(T, D)$ is calculated as follows:

$$M(T, D) = \frac{1}{T \times D} \sum_{t=1}^{T} \sum_{d=1}^{D} \sum^{vec} |z_d^t - \frac{1}{D} \sum_{d=1}^{D} z_d^t|,$$

where $\sum^{vec}$ denotes summation of $n$-dimension vector, $z_d^t$ denotes the object part of the representation extracted from the $d$-th image $\mathcal{I}_d^t$ of the $t$-th object category.

For *integrity*, we reconstruct the image $\widetilde{\mathcal{I}}_d^t$ through swapping the background part of the test image $\mathcal{I}_d^t$ with other background parts. Giving the test images $\{\mathcal{I}_d^t, t \in \{1, 2, 3, ..., T\}, d \in \{1, 2, 3, ..., D\}\}$, the *Integrity Score* $V(T, D)$, which measures the object integrity of reconstructed images, is defined as follows:

$$V(T, D) = \frac{1}{T \times D \times W} \sum_{t=1}^{T} \sum_{d=1}^{D} \sum^{W} \mathcal{M}_d^t |\widetilde{\mathcal{I}}_d^t - \mathcal{I}_d^t|,$$

where $W$ is the pixel number of the image, $\sum_1^W$ denotes summation of image pixel difference value. $\mathcal{M}_d^t$ is the object mask of the test image $\mathcal{I}_d^t$.

## 5 Experiments

In the experiment, One-GORD is compared with the unsupervised method, the semi-supervised method, and the supervised method. Those methods are validated on five datasets qualitatively and quantitatively. What's more, experiments demonstrate the comparative performance in the practical application: classification and image editing.

### 5.1 Implementation Details

**Dataset**  To verify the effectiveness of the proposed One-GORD, we adopt five datasets: SVHN [29], CIFAR-10 [3], COCO [19], Mugshot [11], and mini-ImageNet [24], which are composed of different objects and complex backgrounds. For the COCO dataset, we choose ten object categories (bird, bottle, cat, dog, laptop, truck, tv, tie, sink and book). For the rest of datasets, all the categories are

adopted in the experiment. The training and testing sample numbers are (20000,1000), (20000,1000), (40000,1000), (30000,1000), (10000,1000) for SVHN, CIFAR-10, COCO, Mugshot, and mini-ImageNet, respectively.

**Network architectures**   The encoders and decoders have the same architecture as ResNet[2] [9] . The classifier network is the two-layer MLP with 20 and $N$ neurons for each layer, where $N$ is determined by the category number for each dataset. The Adam algorithm is adopted. The learning rate is set to $0.0005$.

**Parameters settings**   In the experiment, the balance parameters $\tau$, $\alpha$, $\gamma$, $\eta$, $\lambda$ are set to 1, and $\beta$ is set to 10, $\rho$ is set to 1000, and $\delta$ is set to 5. Through large experiments, we find that the crucial parameter are $\beta$, $\rho$ and $\delta$. Tuning $\beta$, $\rho$ and $\delta$ may lead to better performance under the condition that $\tau$, $\alpha$, $\gamma$, $\eta$, $\lambda$ are set to 1.

### 5.2   Qualitative Evaluation

In the qualitative evaluation experiments, our methods are compared with AE, S-AE, DSD [11], MONet [6] and IODINE [12], which are shown in Fig. 2. AE is the basic autoencoder architecture. S-AE is the AE with a classifier, which supervises different parts of latent representation to extract object-related features with object labels. For DSD, the annotated input pairs are generated with the augmented annotated one-samples.

From Fig. 2, we can see that the object swapped images reconstructed by the AE have overlapping features of the two input images. It indicates that the latent representations extracted by AE have mix features of objects and background. In the object swapped images reconstructed by the DSD, the objects have object features of two input images, which demonstrates that the DSD fails in disassembling object features from the background with the same annotated one-samples. For the MONet [6] and IODINE [12], the splitted objects and backgrounds of SVHN dataset are wrong. What's more, they fail on the CIFA-10, COCO and Mugshot, which verifies that the existing unsupervised methods fail in handling real-world images with complicated background. It's noticeable that the corresponding objects are swapped successfully in the results of One-GORD for the above five datasets, which verifies that One-GORD can handle real-world images with complicated background effectively. Even the reconstructed images lost some details, the swapped object and background only contain their corresponding features. In the second row of Fig. 2, the images reconstructed by autoencoder also not achieve perfect reconstructed results. The image reconstruction quality is usually decided by the dataset and network architecture, which will be optimized in our future works.

### 5.3   Quantitative Evaluation

To compare different methods quantitatively, we adopt the *Modularity Score* and *Integrity Score* (Section 4) to measure the disassembling performance of our methods with S-AE, DSD [11], MONet [6] and IODINE [12]. In the experiments, $T$ and $D$ are set to 10 and 100, respectively. We sample 5 kinds of representation length ($\{10, 20, 30, 40, 50\}$) and test all methods in those length setting. Table 1 gives the average modularity score (AMS) and average integrity score (AIS) on the SVHN dataset (the first three rows) and the CIFA-10 dataset (the last two rows).

For the modularity score, One-GORD has the smallest score than other methods, which shows that the disassembled object representations extracted by One-GORD are more similar than other methods' for the images with the same object and different backgrounds. O-G$_s^-$,O-G$_f^-$ and O-G$_o^-$ achieves the larger score than One-GORD on two datasets, which indicates the necessity of One-GORD's each component. What's more, S-AE, DSD, MONet and IODINE achieve larger scores than One-GORD. It means that existing methods can't disassemble object representation effectively, which is in accordance with the reconstructed visual results in Section 5.2.

For the integrity score, MONet and IODINE achieves the larger score than other methods, which verifies that MONet and IODINE fails in disassembling object representation for the real-world image with a complicated background. The average integrity score of S-AE is higher than the score of One-GORD, which shows that the supervision of the object label is not enough for disassembling

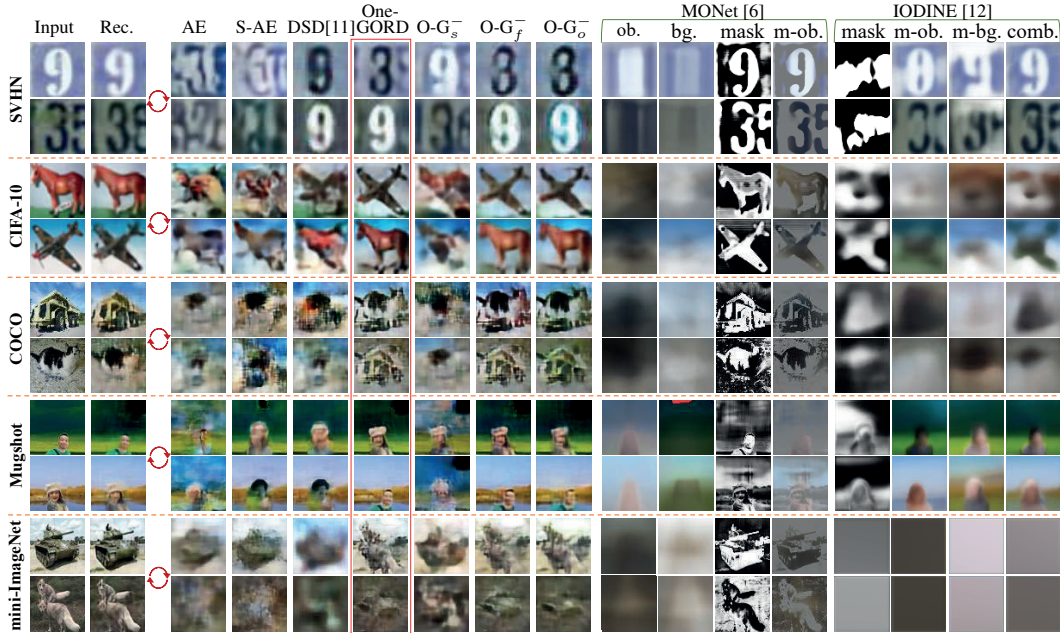

Figure 2: The qualitative results of different methods on five datasets. For each dataset, given two input images, we show the images reconstructed with the object parts swapped representations. For MONet [6] and IODINE [12], we show the splitted objects and backgrounds. 'Rec.' denotes reconstructed results of original images by AE. O-$G_s^-$, O-$G_f^-$ and O-$G_o^-$ denote the One-GORD without supervision module, fuzzy classification, and object reconstruction loss. 'ob.' and 'bg.' denote object and background. 'm-ob.' and 'm-bg.' denote masked object and masked background. 'comb.' denotes the combined result of 'm-ob.' and 'm-bg.'.

Table 1: The Average Modularity Score (AMS) and Average Integrity Score (AIS) in five representation length setting on SVHN (the first three rows) and CIFA-10 (the last two rows) datasets.

| Metric | S-AE | DSD [11] | MONet [6] | IODINE [12] | One-GORD | O-$G_s^-$ | O-$G_f^-$ | O-$G_o^-$ |
|---|---|---|---|---|---|---|---|---|
| AMS | 13.69 | 12.38 | 11.52 | 15.78 | 5.91 | 11.58 | 17.01 | 13.30 |
| AIS | 6.51 | 3.02 | 10.31 | 14.31 | 2.05 | 4.05 | 5.67 | 2.45 |
| AMS | 15.82 | 14.97 | 16.83 | 19.04 | 8.43 | 12.43 | 18.21 | 15.31 |
| AIS | 8.21 | 6.94 | 11.96 | 15.34 | 5.21 | 7.92 | 9.37 | 7.94 |

object features from background integrally. One-GORD achieves the smallest score among all methods, which indicates that the reconstructed objects are more intact than other methods'.

## 5.4 Ablation Study

In the One-GORD, the total loss $\mathcal{L}$ is composed of loss terms from two modules: augmented one-sample supervision module and guided self-supervision module. To verify the necessity of the augmented one-samples, we remove the loss terms in the supervision module, which is denoted as O-$G_s^-$. Meanwhile, we also do the ablation study by removing the fuzzy classification loss $\mathcal{L}_{rec}^z$ and object reconstruction loss $\mathcal{L}_{rec}^o$ from the guided self-supervision module, which are denoted as O-$G_o^-$ and O-$G_f^-$, respectively.

Table 1 gives the average modularity and average integrity scores of One-GORD, O-$G_s^-$, O-$G_f^-$ and O-$G_o^-$ in five representation lengths ($\{10, 20, 30, 40, 50\}$). It's noticeable that O-$G_s^-$ achieves the largest average modularity and integrity scores than O-$G_f^-$, O-$G_o^-$, One-GORD, which demonstrates the necessity of the annotated one-sample. O-$G_o^-$ achieves a relatively high modularity score and a relatively small visual integrity score. The reason is that the object reconstruction loss can effectively

Table 2: The classification performance on SVHN and CIFA-10 (All scores in %).

| *Dataset* | *Metric* | S-AE | DSD [11] | MONet [6] | IODINE [12] | One-GORD | O-G$_s^-$ | O-G$_f^-$ | O-G$_o^-$ |
|---|---|---|---|---|---|---|---|---|---|
| SVHN | C-P | 56.91 | 45.66 | 54.56 | 43.86 | 60.94 | 58.27 | 57.76 | 57.46 |
| | C-R | 57.87 | 46.60 | 54.86 | 42.78 | 59.18 | 58.11 | 58.04 | 57.40 |
| | O-P | 57.27 | 45.20 | 54.67 | 43.89 | 61.47 | 57.60 | 57.93 | 57.17 |
| | O-R | 57.37 | 45.31 | 54.97 | 44.81 | 60.47 | 57.69 | 57.33 | 57.26 |
| CIFA-10 | C-P | 44.93 | 41.27 | 43.59 | 39.43 | 46.23 | 39.61 | 43.17 | 44.21 |
| | C-R | 45.81 | 41.26 | 42.71 | 38.81 | 47.38 | 38.83 | 43.82 | 43.96 |
| | O-P | 45.93 | 41.27 | 43.83 | 37.49 | 46.31 | 40.82 | 41.62 | 42.81 |
| | O-R | 45.93 | 41.26 | 41.86 | 38.26 | 46.73 | 42.96 | 44.21 | 41.97 |

promote the reconstruction quality of the object. However, it affects the modularity of latent representation negatively, which leads to a higher modularity score. Without fuzzy classification, O-G$_f^-$ achieves a higher score than One-GORD, which verifies the effectiveness of the fuzzy classification loss.

Table 2 shows the classification performance of One-GORD, O-G$_s^-$, O-G$_f^-$ and O-G$_o^-$ on SVHN and CIFA-10, respectively. We can see that One-GORD achieves the best classification performance than other methods, which demonstrates that fuzzy classification, object reconstruction loss, and supervision module can enhance the disassembling performance effectively.

## 5.5 Application

As described above, our method can be applied to many machine learning tasks, including image classification, image editing, visual concepts learning, and so on. In this section, we test the performance on two basic applications: image editing and image classification.

For image editing, given one image, objects in the other seven images are swapped into it. The object swapped results are shown in Fig. 3, where we can see that the corresponding objects are successfully swapped for the five datasets. However, there are still some details lost in the reconstructed images, which will be studied in our future work. The benefit of image editing in the latent representation space is that the obscured backgrounds by the objects can be reconstructed well.

For image classification, we compare our methods with other methods on SVHN and CIFA-10 with 1000 test samples. The per-class and overall precision (C-P and O-P) and recall scores (C-R and O-R) are calculated for the above methods, where the average score is taken over all classes and all test samples, respectively. To compare fairly, after obtaining the disassembled representation for each method, we adopt the same linear SVM [1] to train and test the classification performance. The classification performance on SVHN and CIFA-10 are shown in Table 2, respectively . We can see that our method achieves a higher score than other methods, which demonstrates that the object features extracted by our method are more intact and independent. Meanwhile, the O-G$_f^-$ and O-G$_o^-$ achieve the lower scores than other methods, which verifies the effectiveness of the fuzzy classification and object reconstruction loss once again. It's noticeable that even with the supervised label, the S-AE still achieves lower accuracy and recall scores than One-GORD, which indicates that disassembled object representation can effectively improve the classification performance.

## 6 Conclusion

In this paper, we propose the One-GORD, which only requires one annotated sample for each object category to learn disassembled object representation from unannotated images. One-GORD is composed of two modules: the augmented one-sample supervision module and the guided self-supervision module. In the supervision module, we generate some augmented one-samples with data augmentation strategies. Then, the annotated mask and object label supervise the disassembling between the features of the object and background. In the self-supervision module, two self-supervised mechanisms (fuzzy classification and dual swapping) are adopted to generate self-supervised information, which can disassemble object representation of unannotated images with the guidance with annotated one-samples. What's more, we devise two disassembling metrics to measure the modularity of

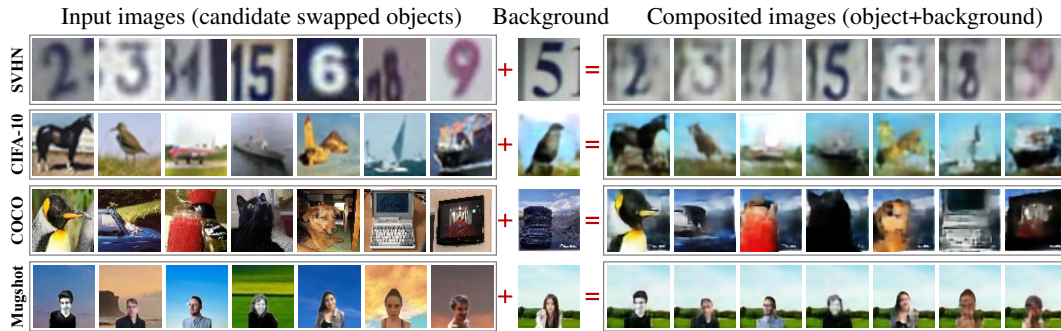

Figure 3: The image editing results on different datasets.

representations and the integrity of images, respectively. A large number of experiments demonstrate that the proposed One-GORD achieve competitive dissembling performance and can handle natural scenes with complicated backgrounds. In future work, we will focus on disassembling objects into different parts and optimizing network architecture to solve the details lost.

## Acknowledgement

This work is supported by National Natural Science Foundation of China (61976186, 62002318), Zhejiang Provincial Science and Technology Project for Public Welfare (LGF21F020020), Programs Supported by Ningbo Natural Science Foundation (202003N4318), the Major Scientific Research Project of Zhejinag Lab (No. 2019KD0AC01) and Alibaba-Zhejiang University Joint Research Institute of Frontier Technologies.

## Broader Impact

This research belongs to the image representation learning area. Positive: the proposed method can be applied to many machine learning tasks, including image editing, image classification, few/zero-shot learning, and visual concepts learning. It supplies a universal tool for other downstream tasks. Negative: the research can be adopted to generate some fake image, which also can be used for malicious purposes.

## Footnotes

[2]https://github.com/cianeastwood/qedr

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
