[Supplementary Material · README.pdf]

Codes and guides of paper: **One-sample Guided Object Representation Disassembling (Paper ID 2171)**

In the 'code.zip' folder, we provide the source codes. Here, we give the full training and testing guides of SVHN experiment.

# Dependencies

```
python3
tensorflow=1.8
pillow
py-opencv
scipy<1.3.0
scikit-learn
```

# SVHN Experiment

We give codes of all experiments on the SVHN datasets in our paper, including AE, S-AE, DSD, One-GORD and its ablation study methods. About MONet and IODINE method, please refer to: https://github.com/baudm/MONet-pytorch and https://github.com/zhixuan-lin/IODINE.

Here, we set 'unitLength=50' as an example. The part length is the length of each part of the representation, which can be set to any integer that is large than 0. In the code file, the part length is denoted as 'unitLength'.

## Datasets

First, Download the dataset.

The dataset generation examples (all datasets are saved in `./npz_datas/` as `npz` format) are given as follows:

1. Run `main_generateSVHN10_train_forOnesample` to generate the **training datasets** for One-GORD and AE.
2. Run `main_generateSVHN10_train_forSAE.py` to generate the **training datasets** for S-AE.
3. Run `main_generateSVHN10_train_forDSD.py` to generate the **training datasets** for DSD.
4. Run `main_generateSVHN10_testWithLabel_forSwapVisual.py` to generate the visual images for image editing.
5. Run `main_generateSVHN10_testWithLabel_forMetrics.py` to generate the **testing datasets** for classification metrics evaluation.
6. Run `main_generateSVHN10_testWithLabel_forModularity.py` and `main_generateSVHN10_testWithLabel_forVisualIntegrity.py` to generate the **testing datasets** for modularity and integrity evaluation.

## Training

After preparing the datasets, we can train the model with `main.py`, which is given in the directory of each method.

One-GORD

```
cd One-GORD\Ours
python main.py
```

Ablation study

```
cd One-GORD\Ours-f # Ours-0 or Ours-s
python main.py
```

AE

```
cd AE
python main.py
```

S-AE

```
cd SAE/SAE_part2_unitLength50
python main.py
```

DSD

```
cd DSD\dual_diaeMnist_unitLeng50
python main.py
```

The intermediate results in training stage will be saved in `./samples/`

## Testing

Now we can use the trained model to do visualization testing! Please choose the best model which can produce the best result in `./samples/` folder. For example, the best reconstructed image is `2000X1head_aux1bg.png`, which means that we have the best model's ckpt when step=2000.

For example, with step=2000, we need modify one line code in the corresponding test file.

```
# change the ckpt number
saved_step = AE.load_fixedNum(inter_num=2000)
```

One-GORD

```
cd One-GORD\Ours
python test_for_SwapVisual.py
```

Ablation study

```
cd One-GORD\Ours-f # Ours-O or Ours-s
python test_for_SwapVisual.py
```

AE

```
cd AE
python test_for_SwapVisual.py
```

S-AE

```
cd SAE/SAE_part2_unitLength50
python test_for_SwapVisual.py
```

DSD

```
cd DSD\dual_diaeMnist_unitLeng50
python test_for_SwapVisual.py
```

The results will be saved in `./VisualImgsResults/`

## Calculate metric scores

We can `cd` to each method directory for calculating the metric scores. Notes: please set the trained model before running scripts. We give One-GORD as the example and the other methods are the same.

One-GORD as example

```
# cd our_work_path
cd One-GORD/Ours # or other method directory
# please set the ckpt which to be loaded from our_work_path
python test_getRepreCodes_forMetrics.py
# calculate classificaiton metrics
python classify_metrics_cal.py
# please set the ckpt which to be loaded from our_work_path
python test_for_VisualIntegrity.py
# calculate visual integrity metrics
python visualIntegrity_metrics_cal.py
# please set the ckpt which to be loaded from our_work_path
python test_for_Modularity.py
# calculate modularity metrics
python modularity_metrics_cal.py
```