[Reviews · NeurIPS 2020]

Review 1

Summary and Contributions: This work proposes a novel scheme to `disassemble' features of an object in an image. To this end, the authors propose to combine a supervised module and an unsupervised module to carry out the disassembling between the object and the background. In the supervised module, the one sample teaches the network to disassemble features of the foreground object from the background. In the unsupervised module, the authors adopt a dual-swapping strategy and a fuzzy classification. Specifically, the fuzzy classification loss constrains the predicted probability to produce only one peaky response across all classes. The author also come up with two metrics to evaluate the performance of the disassembling. The proposed approach works consistently well on four datasets.

Strengths: 1) In my opinion, such a disassembling task using only one sample is novel. The problem setup is practical and may potentially benefit some real-world applications. 2) The fuzzy classification loss is innovative. It is designed for constraining the predicted probability to respond to only one class, in which way self-supervised information for the unlabeled samples is generated. 3) The authors propose two new metrics to evaluate disassembling performance, making the study on the disassembling task thorough and systematic. 4) The authors have released their code, making the result-reproducing effortless.

Weaknesses: 1) More analyses should be given on the performance vs. the length of the representation, which to me seems an important contribution. Does the proposed method lead to stable results with different representation lengths in all datasets? Also, is there a common set of balance parameters for different methods? 2) What will happen if the image contains more than two types of objects? Will the method still work? Discussions should be provided. 3) What is the exact goal of the dual-swap step? Why a dual step is necessary? This should be clarified. 4) How many supervised samples are used in the DSD method? Why are the reconstructed results of MONet [6] and IODINE [12] different from the other methods? Please elaborate.

Correctness: Yes, I think they are correct.

Clarity: Overall, this paper is well written and easy to follow. Yet, I suggest the authors carefully proofread this submission, some typos need to be revised. For examples: Line 4, "unsupervised methods can't .." -> "unsupervised methods cannot ..." Line 62, "Meanwhile, We ..." -> "Meanwhile, we ...".

Relation to Prior Work: Yes, it is clearly discussed.

Reproducibility: Yes

Additional Feedback: [After Rebuttal] After carefully read all reviews and the authors' feedback, I still incline towards acceptance. The authors' feedback basically solves all of my concerns. I still think the authors made a good attempt on representation learning. It may inspire related researches in our community. In addition, for learning disentangled and here proposed 'disassembling' representations from real-world data, we still have a long way to go. Therefore, I hope the authors would include more discussions that appear in their rebuttal in the final (or next) version, such as the limitations in the current stage. It would benefit future studies. Finally, I would like to adjust my score to 8. ----------------------------------------------------------------------------------- 1) In the experiments, the author mentioned that ‘We sample 5 kinds of representation length ({10; 20; 30; 40; 50}) and test all methods in those length setting.’ Again, as mentioned in the Weakness, how does the representation length influence disassembling, and how are such lengths set? 2) Why do the authors choose a two-layer MLP with 20 neurons as the classifier? Does the choice of the classifier affect the performance of the object representation disassembling?


Review 2

Summary and Contributions: The paper outlines a framework for learning to separate single objects from their backgrounds in natural images, and representing those elements in distinct parts of a latent space (which the authors refer to as representation disassembling). This is learned in a semi-supervised setting which requires only single annotated examples from each object category in the dataset. To encourage completeness of the representations, the framework requires reconstructing the pixels from the representations, and that object category can be decoded from the representations on the annotated examples. The framework makes more use from the limited amount of annotated data by augmentation. A key part of this uses the spatial mask in the annotations that separates the object from its background to switch background and objects between (annotated) examples. In addition to the standard reconstruction and classification objectives, the model is trained under a range of additional constraints which encourage the object and background to be encoded in separate parts of the model's latent space. These involve swapping the parts of the representation between pairs of data examples combined with corresponding swaps in the data using the available annotations. The authors present results on a variety of datasets that feature natural images and the required augmentations (such as COCO and CIFAR-10), with both qualitative and quantitive evaluations, comparisons with baselines and model ablations. They also demonstrate uses of their representations for an image editing task, and boosting classifier performance.

Strengths: Key strengths of this work include: - Working toward learning complete representations of scenes which explicitly separate objects and backgrounds. At a high-level, this is a goal that is relevant to computer vision, representation learning and object-oriented RL communities. - Clever use of limited annotations with data augmentation and representational manipulations to encourage the desired separation in the bulk of the unsupervised data. This includes a novel cyclic swapping procedure which is reminiscent of Cycle GAN (which the authors term "dual swapping") which may be promising for opening up approaches with even less supervision. Their results seem to demonstrate a surprisingly data efficient use of single annotated examples for each object category. - Applying the approach to a variety of datasets with natural images, which broadens the scope and utility of this work. - A reasonable body of results are presented to evaluate the method.

Weaknesses: A critical limitation is that this framework seems to be fit to the setting of scenes with one object against a background. This would appear to severely limit the utility of the work for a variety of task settings (including their example task of image editing). The paper does not discuss this limitation or explain how the approach could be extended to work in the multi-object setting. Relatedly, the authors have picked MONet and IODINE as key baselines, both of which are models designed to handle a multi-object setting but haven't been shown to perform well on natural scenes -- in contrast with One-GORD. In addition, those models can learn disentangled feature representations of scene objects. From reading the motivation in the introduction of the paper, this would seem to be a goal here too, but the authors haven't presented any analyses of similar results in their model (beyond their object class classifier).

Correctness: While there are significant issues with the introduction and related works (more on this below), the core claims and methods seem reasonable. However, the paper would benefit from being much clearer in laying out the goals and limitations of the approach, particularly regarding the focus on separating out single objects from backgrounds (as discussed above).

Clarity: There are significant issues with the clarity of the paper, particularly in the introduction and the early part of the methods sections, and the conclusion. In the introduction, (and the conclusion) a key issue is being able to understand what problem is being addressed in the approach, how this method has been developed to tackle those issues and what its limitations are. As part of this it would help to be clear from the start that the goal of this work is to learn representations which distinguish objects and backgrounds. In some cases the paper seems to adopt non-standard or unintuitive terminology, and often doesn't define what is meant until much later. For example, the authors use the term "representation disassembling" throughout the paper, which is new to me. I was unsure what was meant by this term throughout most of the reading until arriving at the Disassembling Metric section. As this term is relied on throughout the discussion -- including heavily in the introduction (which is intermingled with discussions of the widely used term "disentanglement") -- it would help enormously with the clarity of the paper if this was defined early on. e.g. in line 1 of the abstract "The ability to disassemble the features of objects and background", which even primes the reader to expect separation to concern the features of objects, rather than objects from backgrounds. It might be worth considering using alternative, more standard naming (for example something like "scene decomposition" if that indeed aligns with the authors intent). Another example of unusual terminology choice was "integrity", where perhaps something like "reconstruction accuracy" would trigger more directly relevant associations in the reader. Another minor terminology suggestion would be for the "double swapping" process to include something like "cyclic" in its naming. This could help remind the reader that under desired conditions, the outputs should return to how they were. Finally, as the approach itself is unpacked in the introduction and early methods, it is quite hard to follow what the pieces do at a high-level and why they have been introduced. Having better laid out the main goals of the work previously, it would then be helpful if the authors clearly laid out the moving parts of the approach at a high-level and discuss their purpose, etc.

Relation to Prior Work: The related work section needs improvement. The authors have discussed results from unsupervised feature-level disentangling and multi-object scene decomposition. However, this section isn't clear about what is different from those results. e.g. that this work does not tackle feature disentangling at all, and that this approach focuses on single-object natural scenes. There is mention of previous approaches not working with complicated backgrounds, but this is unqualified and no detail is given as to what is meant by this (which could for example be realistic backgrounds from real world images).

Reproducibility: Yes

Additional Feedback: To summarise, as discussed there are significant issues in the writing of the paper in clearly laying out the goals of the approach, its limitations and how it fits in with related work. These could be addressed, and the underlying approach of extracting a lot from limited annotations is interesting. However I also have concerns about the scalability of the approach, given its focus on single objects and backgrounds. The work could be more impactful if the work could be extended to show some applicability into more realistic multi-object scenes. [After Rebuttal] After reading the author's feedback and the other reviews, I am revising my score to 6. I appreciate the author's plans to add more results, including on multi-object scenes and addressing the readability concerns about the paper. Regarding multi-object scenes, I would have found this work even more compelling if their goal was to separate foreground objects from each other, in addition to foreground from background.


Review 3

Summary and Contributions: The authors proposed a new object representation disassembling framework. The proposed approach, through a self-supervised mechanism, requires only one annotated sample from each category to disassemble object representations. The idea of fuzzy classification is quite novel. It adopts the assumption that, objects in the same category use the same class label to generate self-supervised information for disassembling object representations, which makes sense. The author introduced two metrics for evaluating the performance of disassembling. Results on the four datasets look nice and do reflect the desired properties of the disassembled representations.

Strengths: The task studied in this paper, which is to disassemble object and background, is very interesting. The method is well designed and makes sense. The framework smoothly integrates two key components: the one-sample guidance and the self-supervised mechanism. The idea of fuzzy classification is novel. It is a smart way to inject self-supervision into the object representation learning. The author proposed two metrics for evaluating the proposed tasks, and have demonstrated promising results on four datasets, showing that object representations are disassembled via their approach.

Weaknesses: My questions are as follows. -- There are quite some balance parameters in the total loss. It might be time-consuming to find the correct the parameter setting, no? How robust is the method to these parameter settings? In other words, will the method deliver reasonable disassembling results with different balance parameters? -- Will the proposed method still work for images containing multiple objects or multiple categories of objects? -- Why only ten object categories are chosen from the COCO dataset? What is the rationale behind?

Correctness: The proposed method is presented clearly. It is correct to my best knowledge. Four datasets are adopted to verify the effectiveness of the proposed method, which is convincing.

Clarity: Yes. The paper is well organized and written.

Relation to Prior Work: Yes

Reproducibility: Yes

Additional Feedback: -- The reconstructed objects in the editing results of Fig.3 seem blurred. Did the author analyze the potential reasons leading to the blurry results? Will more annotated samples help in this case? -- Network architecture plays a vital role on the reconstructed images' visual results. Did the author try any other network architecture? -- Does the fuzzy classification have a connection with ‘fuzzy logic’? Is there any existing technique related to the proposed fuzzy classification? If yes. Please distinguish the proposed one with the existing one. -- It seems that the image size for each dataset is not provided -- Yes, the code is supplied, but still it would be good to have all such details in the manuscript. -- More discussions on the length of representations should be added. How does the representation length affect the final disassembled results? For each dataset, how to find the best representation length? ------------After Rbuttal---------------------- The authors have addressed my concerns well, and also I appreciate that they have released the code. So I prefer to accept this paper.


Review 4

Summary and Contributions: This paper proposes a novel method (or an extension to DSD) to learn disassembled object representation from unannotated images, provided one annotated sample for each object category. Experiments on several datasets demonstrate that the features learned by their method are better w.r.t discriminative power and visual consistency.

Strengths: The work proposes several components to increase the discriminative power of features and validates their effectiveness by comprehensive experiments on multiple datasets. Since there is a trend to use fewer labels to achieve more and better results, the work can serve as a good baseline, although it is still far from real applications.

Weaknesses: 1. Some important details of dataset are missing. For the COCO dataset, I think the authors either use some specific version from others or crop pathces containing objects and resize them to a fixed size. If it is the case, what is the input resolution? 2. It will be nice to show the sensitivy of those important hyper-parameters mentioned in L200. 3. There seem to exist a limitation of the method. Since the method relies on reconstruction, it may be difficult to deal with larger images and more diverse objects. Can the authors show results on ImageNet (e.g., mini-ImageNet)? 4. Can the authors explain L281-282 with more details? It seems non-intuitive that supervised methods are worser than semi-supervised methods under the same condition. 5. (Minor) Some details of baselines are missing. Although I agree that unsupervised baselines, like MoNet and IODINE, can not deal with complex backgrounds (even with modifications I will mention later), I would like to discuss about a fairer comparison. Those unsupervised, VAE-based methods leverage Spatial Broadcast Network as decoders to perform well on toy datasets. The capacity of this decoder favors certain complexity, and unsurprisingly fails on complex backgrounds. Thus, do the authors use ResNet as the decoder of MoNet or IODINE and tune thier hyper-parameters? 6. (Minor) Most equations relevant to classification in Sec 3 look strange to me. It seems that the authors express the negative log likelihood in a way not familiar to me.

Correctness: Yes.

Clarity: Writing is fluent and clear, but some equations can be improved.

Relation to Prior Work: Yes.

Reproducibility: Yes

Additional Feedback: Where is $\lambda_{cla}^{d}$ defined? line 251: largest -> lowest? ---- The rebuttal addressed my major concerns. I will stick to my original rating for accepting the work.

[Author Response · NeurIPS 2020]

**One-sample Guided Object Representation Disassembling** (Paper ID 2171)

We sincerely thank all reviewers for the constructive comments. We provide short responses here due to the page limit.

─────────────────────────── **To Reviewer #1** ───────────────────────────

**Q1: a) How does the representation length influence disassembling? b) Is there a common set of parameters?**

**A1:** Smaller length yields better representation disassembling, while larger length leads to better image reconstruction.

For images with complex scenes, the representation length should be set larger. b) Yes. They are given in Line 199.

**Q2: Will the method still work on multiple object image? Q3: What is the exact goal of the dual-swap step?**

**A2:** In this case, features of multiple objects will be disassembled into the same part. Visual results are shown in Fig. I.

**A3:** The dual-swap step is devised for generating self-supervised information for the unannotated samples.

**Q4: a) Number of supervised samples in DSD. b) Why are the results of MONet [6] and IODINE [12] different?**

**A4:** a) Around $5,000$ samples augmented with annotated one-samples are used. b) MONet [6] and IODINE [12] learn

object representations and corresponding masks. The fixed masks lead to the non-swappable defects of [6,12].

**Q5: a) Why choosing two-layer MLP as classifier? b) Does the classifier affect the disassembling effect?**

**A5:** a) It is a simple classifier and trainable by gradient descent. b) Yes. Simpler but not too simple classifiers better

reflect the properties of the representation, which favors the disassembling.

─────────────────────────── **To Reviewer #2** ───────────────────────────

**Q1: Extension to multi-object setting. Q2: Disentangling features of scene objects seems a goal of the method.**

**A1:** Thanks for the comment. The method is indeed able to handle images with multiple objects, in which case features

of multiple objects will be extracted together and denoted by the same part of the representation. Fig. I below shows the

visual results of multi-object samples in COCO and miniImageNet (a newly added dataset). More results will be added

to the revision. **A2**: In fact, disentangling features of objects is not our goal. Rather, our method aims to disassemble

features between one-sample guided objects and the background. Features of guided objects are denoted by one part of

the representation, while features of unguided objects, which are treated as the background, are denoted using the

remaining part of the representation. We will clarify this in the revision.

**Q3: Writing issues: unclarity of the introduction and conclusion; non-standard or unintuitive terminologies.**

**A3:** Thanks for the nice suggestions. We will clearly list the main goals in introduction and amend terminologies.

**Q4: Improving related works. Q5: Meaning of prior methods not working on complicated backgrounds.**

**A4:** Thanks. We will revise this part and provide clearer distinctions between our method and prior ones. **A5:** We
meant that, prior methods like MONet [6] work on synthetic images only but not real-world ones. We will clarify this.

Figure I: Visual results reconstructed with original or object-swapped representations on multi-object images.

─────────────────────────── **To Reviewer #3** ───────────────────────────

**Q1: How robust is the method to the parameters? Q2: Will the method still work for multi-object images?**

**A1:** Please refer to **A1(b)** of **R#4**. **A2:** Yes. Fig. I shows the results on multi-object images.

**Q3: Why choosing only ten categories from COCO? Q4: The blurred results in Fig. 3. Other architectures?**

**A3:** The sample numbers in these ten categories are balanced. **A4:** Small representation length favors disassembling

but harms image reconstruction, leading to the blurred results. We tested the method with other architectures, yet results

showed that changing architecture barely helps. More labeled samples help improve the blurred results.

**Q5: Connection with 'fuzzy logic'? Q6: Image size? Q7: How to find the best representation length?**

**A5:** No. **A6:** SVHN:$32 \times 32$; Others: $64 \times 64$. **A7:** Setting larger representation length for a more complicated image.

─────────────────────────── **To Reviewer #4** ───────────────────────────

**Q1: Are samples of COCO cropped to patches and resized? Q2: The sensitivity of hyper-parameters.**

**A1**: Thanks. We only resized samples to $32 \times 32$ (SVHN) or $64 \times 64$ (others), but did not crop them. **A2**: The vital

parameters $(\beta, \rho)$ and $\delta$ respectively control the disassembling on annotated and unannotated samples. The (AMS,

AIS) for setting $(\beta, \rho, \delta)$ to be $(1, 100, 5)$, $(10, 1000, 5)$, and $(10, 1000, 50)$ are $(14.32, 7.83)$, $(13.69, 6.51)$, and

$(13.52, 6.31)$, respectively. Reasonably large $\beta$ and $\rho$ favor disassembling. More results will be added to the revision.

**Q3: Showing results on miniImageNet. Q4: Classification equations are expressed in an unfamiliar way.**

**A3:** Fig. I(b) shows visual results; the AMS and AIS scores for (S-AE, DSD, MONet, IODINE, Ours) are respectively

$(14.45, 13.36, 10.43, 17.49, 6.98)$ and $(7.81, 5.76, 9.82, 18.93, 4.03)$. **Q4**: Our loss requires computing each value in

the predicted class vector. To be consistent, therefore, equations are written in an expanded form. We will clarify this.

**Q5: Explaining why supervised methods (S-AE) are worse than semi-supervised methods (One-GORD).**

**A5:** Thanks. With only labels and a two-layer MLP classifier, S-AE is not able to effectively extract the entire

discriminant features. However, dual-swap and object reconstruction of our method favor representation disassembling.

**Q6: Try ResNet as the decoder of MoNet or IODINE to deal with complex backgrounds. Q7: $\lambda_{cla}^d$? and typos.**

**A6:** Thanks. Still, the modified MoNet and IODINE do not work. One reason is that the Gaussian-distribution

assumption facilitates decomposition but limits the representation's capacity. **A7:** It should be $\lambda_{rec}^d$. We will fix typos.

[Meta-Review · NeurIPS 2020]

Four knowledgeable referees support accept and I accept. We encourage and expect the authors to incorporate the reviewers' suggestions for improving the paper. In particular, please address concerns regarding readability, and please add more results including on multi-object which will improve the work.